# The Chemical Ecology of Elephants: 21st Century Additions to Our Understanding and Future Outlooks

**DOI:** 10.3390/ani11102860

**Published:** 2021-09-30

**Authors:** Bruce A. Schulte, Chase A. LaDue

**Affiliations:** 1Department of Biology, Western Kentucky University, Bowling Green, KY 42101, USA; 2Department of Environmental Science and Policy, George Mason University, Fairfax, VA 22030, USA; cladue@gmu.edu

**Keywords:** chemical signals, conservation, husbandry, interspecific interactions, mate assessment, musth, pheromone, reproduction

## Abstract

**Simple Summary:**

Among all taxa, messages transmitted via chemical signals are the oldest and most universal. For Asian and African elephants, odors convey information between individuals, and between elephants and their environment. Pheromones are chemical signals used within a species and while thousands of pheromones have been identified for insects, only a few dozen have been elucidated in mammals and other vertebrates. Amazingly, two pheromones are known for Asian elephants: one signals receptivity in females and the other a heightened reproductive state in males. The elephant trunk serves numerous functions including detecting airborne odors and transporting chemicals in substrates such as urine to be detected by multiple sensory systems. Obvious trunk behaviors provide clear means to assess the interest of elephants in scent sources. Thus, elephants can serve as a model system for investigating chemical signaling. Prior to the 21st century, research on elephant chemical signaling focused on within species communication. In the 21st century, these studies expanded with increasing fieldwork. Studies also revealed the use of odors to detect threats and forage for food. Chemical signaling in elephants remains a bouquet for further exploration with promising applications for the conservation of wild elephants and the management of elephants in human care.

**Abstract:**

Chemical signals are the oldest and most ubiquitous means of mediating intra- and interspecific interactions. The three extant species of elephants, the Asian elephant and the two African species, savanna and forest share sociobiological patterns in which chemical signals play a vital role. Elephants emit secretions and excretions and display behaviors that reveal the importance of odors in their interactions. In this review, we begin with a brief introduction of research in elephant chemical ecology leading up to the 21st century, and then we summarize the body of work that has built upon it and occurred in the last c. 20 years. The 21st century has expanded our understanding on elephant chemical ecology, revealing their use of odors to detect potential threats and make dietary choices. Furthermore, complementary in situ and ex situ studies have allowed the careful observations of captive elephants to be extended to fieldwork involving their wild counterparts. While important advances have been made in the 21st century, further work should investigate the roles of chemical signaling in elephants and how these signals interact with other sensory modalities. All three elephant species are threatened with extinction, and we suggest that chemical ecology can be applied for targeted conservation efforts.

## 1. Introduction

Field studies from all three elephant species (Asian elephants, *Elephas maximus*; African savanna elephants, *Loxodonta africana*; and African forest elephants, *L. cyclotis*) indicate common sociobiological patterns [1,2]. Females are at the center of elephant social structures in which they exhibit philopatry and retain close bonds with their natal groups for life [3,4,5,6]. These groups are multileveled with older females exhibiting dominance over younger ones, and social organization is maintained by fission-fusion processes [2,4,7,8]. Upon reaching sexual maturity, male elephants remain on the periphery of these female-centered groups; younger males often form social relationships with each other before becoming more solitary at their sexual peak [9,10,11,12]. Male and female elephants are generally spatially segregated, except during periods of reproduction when females are in estrus and/or males are in a heightened reproductive period called “musth” [13,14,15]. While musth occurs regularly (although asynchronously between males), female elephants are only in estrus for several days every four to five years due to long gestation and weaning periods [16,17,18,19]. Therefore, synchronized breeding is critical, and elephants must navigate complex social environments to reproduce. Elephants rely on a milieu of chemical signals to announce their reproductive intent and to ascertain the receptivity of conspecifics. All three species are megaherbivores with broad yet selective palates [1,20,21]. Because of their size, elephants are considered largely free from predation. Nevertheless, large carnivores can be a threat to young, sick, and old elephants, and humans have hunted elephants and their relatives for millennia [22]. Thus, elephants like other species are likely to use chemical signals in food selection and predator detection.

By the end of the 20th century, we had a reasonable understanding of the importance chemical signals play in elephant society. A deeper knowledge existed about chemical signaling in Asian elephants, but the commonality of behaviors, similarities in estrus and musth, and general social dynamics of the species supported hypotheses that African elephants would use chemical signals and perhaps specific pheromones for inter- and intra-sexual communication. More work was needed in understanding the mechanisms of pheromone synthesis and reception. The degree of chemoreception sensitivity by elephants remained to be explored. Of course, chemical signals do not work in isolation. Elephants have excellent hearing and sensitive touch capabilities as well as reasonable eyesight. These senses are covered elsewhere in this issue, so they will not be reviewed here. Nevertheless, we realize the need for research on the interactions across the senses in both perception and signaling [23].

Our enhanced understanding on the importance of chemical signaling to elephants has since prompted investigations into the potential applications in science, husbandry, and management. We first give a background on elephant chemical ecology up to the 21st century; earlier reviews cover these findings in greater depth. Here, we strive to update our knowledge of elephant chemical ecology with research that has occurred over the past twenty years for both pure and applied purposes. Because so little work has been conducted on the chemical ecology of forest elephants, the use of “African elephants” throughout this review relates only to the savanna species (*L. africana*).

## 2. Background up to Turn of 21st Century

By the turn of the current century, intraspecific chemical signaling by Asian and African elephants had received considerable attention (e.g., reviews included [24,25,26,27,28]). One of the critical findings up to this time was the determination that African elephants exhibited musth similar to what had been known for centuries by Asian elephants [14,15,29]. Males of both species dribble urine, secrete an odorous, viscous fluid from the temporal gland (i.e., temporal gland secretions, TGS), can act aggressively, and focus their energies on locating receptive females (reviewed in [23,30,31,32]). Common aspects of musth in behavior and apparent function supported investigating other similarities in the chemical ecology of Asian and African elephants. A second landmark discovery was the identification of the pre-ovulatory pheromone, (Z)-7-dodecen-1-yl (Z7-12:Ac) in the Asian elephant [33,34]. The compound is common in lepidopteran chemical signals, revealing an intriguing example of convergent evolution. With only a few known pheromones in mammals at the time, the identification of a single compound that tracks the receptivity of a female Asian elephant bordered on ground-breaking. Chemical signals appeared to be mediating intersexual elephant reproductive behavior with musth excretions and secretions attractive to females, and at least in Asian elephants, a urinary signal that attracts males.

Chemical signals also play a role for interactions within the sexes. The chemical milieu of musth changes over its duration in Asian elephants with distinct stages [31,35]. In captivity, male Asian elephants distinguish musth and nonmusth urine [36]. African elephants display not only similar behaviors but also hormonal profiles and compounds from TGS comparable to Asian elephants in musth [37]. Males, especially younger ones, may avoid musth males or their secretions [30,38,39]. In Asian and African elephants, the chemistry of TGS, breath, and urine before and during musth was explored with single compounds (e.g., 2-butanone as an indicator of impending musth) or types of compounds (e.g., ketones, alcohols, and aldehydes) as candidates for chemical signals in one or both species [26,40]. Females also respond to chemicals released or emitted by same sex conspecifics (e.g., [41]). Before the 21st century, no compounds from these sources were isolated as chemical signals. Because the secretions and excretions from elephants are so chemically complex, the possibility that chemical signals are composed of blends or bouquets rather than single compounds also was considered likely.

In addition to the behavior and chemistry of these inter- and intra-sexual signals, the anatomy of the structures and physiology of the senders and receivers also had been examined [26,27,30,35]. The elephant trunk serves to sample volatile compounds in the air and chemicals in a liquid medium such as TGS and urine [42]. The African elephant trunk terminates in dorsal and ventral tips, while the Asian trunk exhibits only the dorsal. The tips house sensory vellus vibrissae that likely play tactile and chemosensory roles [43]. Mucus drips from the trunk orifice and proteins can bind with chemicals in liquids such as urine [26]. These anatomical features provide the structural foundation for the observed trunk behaviors.

The availability of discrete, obvious chemosensory behaviors by elephants makes them excellent models for studies on chemical signaling (Figure 1). The trunk can hover over a sample or be lifted to sniff the air, but it also can check a liquid by contact with the trunk tip or even flatten onto the substrate (a place behavior). Once the substrate is contacted, the trunk can transfer chemicals to the openings of the vomeronasal organ in the roof of its mouth in a behavior known as flehmen, which can serve as an indicator of male interest in a female or her urine [44]. For example, in a study with captive Asian elephants, the chemosensory responses of females to male urine depended on the estrous condition of the female and the stage of musth/nonmusth of the male (as determined by serum progesterone and testosterone, respectively) [45]. 

Compared to intraspecific signaling, the study of interspecific chemical signals and odors used by elephants has received less attention. Elephants are megaherbivore browsers and grazers with a broad, yet selective diet in that while they can feed on numerous plant species, they tend to prefer relatively few [21,47,48,49,50,51,52]. From these studies and others, elephant selectivity is based upon plant quality and the degree of plant chemical defenses. Similar to carnivores detecting prey via scent, elephants use the chemical composition of plants to make diet choices, but at the time further work was needed to understand the bases of such choices. Because of their large size as adults, elephants are relatively free from predation, except when elderly or ill. Young, smaller individuals also are susceptible to predation, especially by lions (*Panthera leo*) but also by other predators such as tigers (*P. tigris*), crocodiles (*Crocodylus* spp.), and possibly hyenas (*Crocuta* and *Hyaena* spp.) [53]. Humans have posed the greatest threat to elephants both through direct killing and indirect effects that include climate change and habitat modification [54]. The extent to which odors play a role in the detection of predators or human threats also needed further investigation.

## 3. 21st Century Gains in Understanding

### 3.1. Introduction

The importance of chemical sending and receiving to elephants continues to be solidified in the new century. One of the most powerful recent discoveries in this vein determined that the African elephant genome contained the largest repertoire of olfactory receptor (OR) genes (some 2000) of the 13 placental mammals investigated [55]. When intact gene clusters were determined, elephants had a similar quantity of clusters, but with more genes per cluster, compared to mice and humans. The authors conjectured that the number of OR genes correlates with olfactory discriminatory ability. In a study along such lines with Asian elephants, trained individuals could discriminate among 11 aliphatic acetic esters [56]. The compounds used had been tested with other species, and elephants showed a similar or faster rate of learning to discriminate. In addition, elephants performed better at these olfactory than comparable visual trials. In further exploration of their olfactory abilities, an additional homologous series of aliphatic compounds were tested along with enantiomeric pairs [57]. Elephants were capable of such discrimination within like pairs, and a ten-fold reduction in concentration did not affect their ability. African elephants have been trained to successfully identify trinitrotoluene (TNT) used in land mines [58]. Thus, compounds that may have intra- or inter-specific meaning and even those that likely have no evolutionary context can be perceived and discriminated by elephants.

The anatomy of the chemoreceptive sensory system has received somewhat less attention than the behavioral aspects. The main olfactory bulb in elephants is similar to other mammals with its large size being a notable feature [59]. Interestingly, despite the prevalence of vomerolfactory behavior (i.e., flehmen responses) exhibited by elephants, no accessory olfactory bulb has been located as is found in many mammals [60,61]. The VNO develops in the womb and appears fully functional in newborns, supporting the observation of flehmen-like behaviors in young calf elephants [62,63]. Two decades ago, interdigital glands were identified in Asian elephants that are akin to human eccrine glands [64,65]. We still do not understand the function of these glands and if they contribute to chemical signaling by elephants. Another intriguing finding that has potential for chemical signaling is the antimicrobial properties of TGS [66]. While this could have evolved as a defensive mechanism against infection, the possibility exists that the microbes play a role in chemical signaling as the TGS streams down the cheek and is rubbed on objects in the environment.

### 3.2. Intraspecific Chemical Signaling and Capabilities

Chemical signaling plays an important role both within and between the sexes in elephants (Figure 2). We illustrate some of the landmark discoveries in this section.

In the new century, perhaps the most profound contribution recently was the identification of the compound frontalin (1,5-dimethyl-6,8-dioxabicyclo[3.2.1]octane) as a musth pheromone in the TGS of Asian elephants [67]. Frontalin is not contained in the temporal secretions of young males; these males release a sweet, honey-like TGS [68]. As males age, frontalin levels increase and the ratio of the two enantiomeric forms changes from a greater proportion of the (+) [1R,5S] enantiomer to an almost racemic mixture in older, mid-musth males [69]. Female Asian elephants in their follicular phase distinguish among ratios and concentration of frontalin [70], but this line of investigation was never completed (Dr. L.E.L. Rasmussen was conducting this investigation when she passed away). A study by LaDue et al. [46] used a range of concentrations of racemic frontalin in bioassays with male and female Asian elephants at zoological facilities throughout North America. Older males that had undergone musth were more responsive than younger males who had not been through musth. Similarly, older, sexually experienced females also were more responsive than their younger counterparts. The roles of frontalin still require further investigation; the compound is not only present in TGS but also elephant blood, breath, and urine [30,71].

Research on pheromones in African elephants used the progress with Asian elephants as a starting point. Chemical analyses of urinary volatiles from captive and wild African elephant males revealed similarities across location and provided prospective signals of musth [72]. The Asian musth pheromone, frontalin, also occurs in the ovulatory urine of African elephants [73]. However, bioassays of frontalin in water with wild elephants in Tanzania elicited no heightened responses compared to a vanilla extract control [74]. Additional study suggested that microbes in these secretions may play a role in the signals of musth [75,76,77]. Similar to Asian elephants, hormones, behavior, secretions, and excretions show complementary alterations in African male elephants during musth [78,79]. In general, hormones alter gene expression and cellular function, the effects of which are often observable through behavior. Yet, we also know that the environment (e.g., interactions with conspecifics) can modify behavior that affects hormonal activity. In this regard, we still need a better understanding of the interaction between hormones and intraspecific behavior in elephants.

In situ and ex situ studies with African elephants pursued pure and applied scientific objectives (e.g., [80,81]). Such studies led to an expanded chemosensory ethogram, incorporating not only the main chemosensory behaviors of sniff, check, place, and flehmen [45] but also accessory trunk behaviors [82]. While the displays of main and accessory chemosensory behaviors are not limited to intraspecific communication, their correlation was elucidated through such studies. Using these behaviors, the chemosensory behavior of elephants in response to conspecific urine and feces from two populations—one in Tanzania and the other in South Africa—showed similar patterns across age and sex classes, although somewhat different patterns emerged developmentally between the sexes across the populations [80,83]. Interestingly, chemosensory response patterns in response to luteal and estrous conspecific urine were similar by male African elephants housed at facilities in North America to males in the wild [84]. In the South African population at Addo Elephant National Park, males expressed chemosensory behaviors to a greater extent than females, thus appearing more reliant than females on signals emanating from conspecific feces and urine [85,86]. However, more work on female elephant olfaction needs to be conducted including the role of familiarity (e.g., see [87]) and the possibility of estrous synchrony mediated through chemical signals [88]. Together these findings support the generality of chemosensory behavior at least in African savanna elephants.

Following the discovery of the Asian elephant estrous and musth pheromones, studies were conducted to determine if such signals existed in African elephants. In bioassays at zoological facilities, male African elephants distinguished estrous status from urinary signals [84], yet even with the advent of new techniques that provided candidate compounds [73,89,90,91], a pre-ovulatory pheromone has not been identified. Like Asian elephant females, African elephant females can distinguish the estrous stage of conspecifics but unlike Asian elephants not through urinary signals [92]. In a study of male African elephants at zoological facilities, males performed significantly fewer flehmen behaviors to musth than nonmusth urine but did not show differential responses to urine from early and late stages of musth [93]. In addition, the responding elephants did not show different rates of olfactory behaviors, only flehmen that activates the vomeronasal system. Thus, behavioral studies support the existence of chemical signals—perhaps single compound pheromones—as indicators of estrus and musth in African elephant, but chemical identification and verification is lacking at the present. African elephant intraspecific signals may not be single compounds, but it also seems likely that they are not overly complicated chemical images [94].

While we have been aware of elephants’ abilities to detect intraspecific signals in breath, TGS, and urine, other potential primary or secondary sources of chemical signals are just starting or have yet to be investigated. Male Asian elephants exhibit greater interest in dung from pre- (follicular) compared to post-ovulatory (luteal) phase females [95]. African elephants use olfactory cues (e.g., urine-soaked earth) along pathways and around waterholes potentially to determine the identity and location of conspecifics [86,96,97,98]. Other secretions that potentially release pheromones for intraspecific signaling include saliva, mucus, and fluid from the ears and interdigital glands [24,99]. In particular, saliva may be a potentially rich source of pheromones (or hormones functionally serving as pheromones), as has been observed in other mammals [100,101,102]. The existence of salivary pheromones could explain trunk-to-mouth exchanges frequently observed between elephants [103,104]. Elephants also exhibit a variety of chemosensory behaviors when examining a dead conspecific, although we have little understanding of what information they might be gathering [86,105,106,107]. The ubiquity of intraspecific chemical signaling by elephants challenges our imagination as to the types of information they are gathering. As stated above, a better understanding of their natural products chemistry would help along with further work on the role of microbes [76,108] and anatomy. All these avenues of future research could inform us on not only intraspecific but also interspecific chemical signaling.

### 3.3. Interspecific Chemical Ecology

Animals should be expected to use chemical information from their predators to exhibit a wide range of appropriate responses based on where, when, and how the predator cue is perceived [109]. Except for humans and large cats, elephants are largely free from the danger of predators; in many cases, humans pose the biggest threat. African elephants distinguished between clothing worn by Maasai and Kamba tribesmen using olfactory and visual cues [110]. Their response to the Maasai clothing was more aggressive, perhaps reflective of the increased threat by this tribe in which a rite of passage for men involves spearing an elephant. Lions and tigers are known to prey on elephants where they co-occur, especially calves [53,111,112], and elephants can distinguish the sounds of these predators from controls, responding in a way indicative of delineating degree of threat [113,114]. Using dung from lion and other cats or animals not known to regularly prey on elephants as well as chemicals unique to lion dung, five semi-tame African elephants in South Africa responded in a manner indicative of greatest threat from the lion samples [115]. Interestingly, in this same study, elephant responses did not differentiate between lion and cheetah (*Acinonyx jubatus*), yet the latter would never prey on elephants. Further work is needed to determine if elephants are generalizing across large cats, large carnivores, carnivores in general, or some other categorization. For African elephants, perceived threat can affect movements and grouping behavior [7,113,116], and chemical cues are likely indicators of such threat.

In addition to detecting potential predators, the query arises as to what extent elephants may use odors to inform foraging decisions. One concern for elephants while foraging is the presence of bees. Elephants foraging on trees with hives in them may disrupt the hive, resulting in angry bees attacking them. The sound of angrily buzzing bees evokes alarm calls and avoid behavior in elephants [105]. The smell of honey is associated with bees, and honey odor couples with bee buzzing sounds may indicate danger to elephants [106]. The multimodal signal may be more meaningful and thus resistant to habituation compared to single modality signals [107]. In addition, plant secondary metabolites (PSMs) have been proposed to have evolved in part to reduce mammalian herbivory [117,118], although the relationship between types of herbivory and chemical responses can be complex [119,120]. Elephant foraging behavior has received attention regularly (e.g., [121,122,123,124,125,126,127,128,129,130]), with African elephants classified as mixed-feeders demonstrating selectivity in species and plant parts [131,132] that is influenced by nutritional value, the palatability:tannin ratio, and levels of plant toxins ([133] and citations therein). A study conducted with handled elephants in South Africa showed that foraging choices were based on plant volatiles, especially monoterpenes, that were correlated with PSM content [134,135]. Because male and female African elephants differ in their feeding patterns [121], it would be interesting to examine intersexual differences in preferences to plant volatiles. Asian elephants can use odor but not sound to locate hidden food [136], and they can assess the quantity of a single food type through olfaction [137]. Recently, African elephants were shown to detect olfactory cues associated with water [138]. The body of work to date suggests that odors play an important role in elephant foraging decisions, but more research is needed.

### 3.4. Conservation and Management Applications of Chemical Ecology of Elephants

All three elephant species are threatened with extinction [139,140,141], and many in situ and ex situ elephant populations are intensely managed to control dynamics such as movement, foraging patterns, health, and reproduction that maximize sustainability. The use of olfaction for intraspecific signaling, threat detection, and foraging suggests that a push-pull approach [142] could be useful for elephant management [143,144] (Figure 3).

The ex situ propagation of elephants may be vital for the long-term viability of *E. maximus* and *L. africana*, but even many established captive populations are currently unsustainable [1,145,146]. Therefore, elephant husbandry has been an ongoing area of study with interest in a variety of topics relevant to captive breeding efforts including health, welfare, and enrichment [147,148,149,150,151]. For most terrestrial mammals, the detection of odors is a major means of evaluating their environment and communicating with conspecifics [152], and thus scents have a high potential for enhancing welfare and providing enrichment [153]. However, the selection, implementation, and assessment of scents related to animal well-being can be difficult and requires additional research [154]. The indiscriminate application of chemical signals could result in sensory habituation to these odors, decreasing their efficacy in captive applications. Therefore, a goal-centered approach to olfactory enrichment is necessary, incorporating well-defined behavioral outcomes that indicate success or that suggest modifications are necessary [155,156]. For instance, because many captive elephants commonly exhibit stereotypies and inactivity [157], zoo elephant managers may wish to use pheromones to stimulate activity and increase behavioral diversity. The functionality of these pheromones is sex- and age-specific, so their application should be targeted. Indeed, a recent study of captive Asian elephants showed that while both sex pheromones can effectively increase behavioral diversity and activity, the degree to which these compounds function is influenced by sex, sexual experience, age, and social access [158]. One of the benefits of using pheromones is their inherent biological relevance; thus, we propose that they would be resistant to habituation. Behavioral and physical health is often important in determining reproductive success [159,160]. The ex situ propagation of other threatened mammals (e.g., the giant panda, *Ailuropoda melanoleuca*) has been bolstered by using chemical signals to stimulate reproductive function and activity [161]. Because of the unique reproductive biology of elephants and the artificial social environments of many captive elephants, sex pheromones may also hold promise in enhancing captive breeding efforts in these species. More experimental work is needed to assess how chemical signals could benefit elephant husbandry and captive sustainability.

In the wild, elephant conservation is complicated by the issues of illegal take by humans and competition with humans for limited resources in the forms of habitat, water, and food [162,163,164,165,166]. Chemical ecology has not played a major role in the issue of poaching, although the chemical composition of poached ivory can be used to trace the date of death [167] and DNA analysis helps to isolate location [168]. However, for mitigating human–wildlife conflicts (HWC) including human–elephant conflict (HEC), chemical ecology has been an important contributor [143]. Human–elephant interactions (HEI) can turn into conflict when the two species vie for a common, limited resource such as space, food, or water. HEC may not be an issue of too many elephants or people [169], but rather their distribution when resources are limited (e.g., water during a drought) or at a premium (e.g., crops approaching harvest). At the population level, carrying capacities for elephants have been estimated by considering the negative effects of PSMs on elephant foraging [170]. Intraspecific chemical signals are posited as honest means of communication and thus have potential for modifying the behavior of particular categories of elephants (e.g., adult males or females) in predictable ways [80]. To date, only pheromones for Asian elephants have been identified. Yet, even with Asian elephants, application of the pheromones in the field for conservation purposes has not been conducted in part because the compounds are not readily available, can be somewhat expensive, and a practical means of dispensing has not been formulated. Fortunately, research on the responses of African elephants to bee pheromones [171] has used SPLAT^®^ (Specialized Pheromone and Lure Application Technology) formulated to reduce insect pests [172] in order to dispense the bee pheromones. This matrix might be useful for dispensing the Asian elephant pheromones as well.

In addition to bee pheromones, other odors have been used to deter elephants from human habitations and crops. One avenue is to grow aromatic plants that elephants avoid [173,174]. A problem so far with this approach is that aromatic plants often have medicinal or spice uses but do not provide sustenance. In contrast, sustenance crops such as rice and maize are not highly aromatic. Thus, aromatic plants may be consumed less by elephants, but they may not enhance food security for people. Chili peppers (*Capsicum* spp. or other hot chilis) have been used for as a deterrent in a variety of forms, including growing chilis around sustenance crops, making a motor-oil based extract and creating a cloth and rope fence, by burning, and via gas dispensers (e.g., see these papers and references therein: [175,176,177,178,179,180]). A recent study used high performance liquid chromatography coupled with mass spectrometry to quantify the two major active ingredients, capsaicin and dihydrocapsaicin, [181]. Success using chilis to protect crops has varied widely. The active ingredients are capsaicinods that activate the trigeminal nerve in the elephant trunk, leading to an aversive reaction. The reasons for variable success are potentially many. Further work is needed to examine the responses of elephants to known concentrations of the capsaicinoids. Predatory odors in the form of lion dung and its major, signatory chemical components evoked aversive behavior in handled elephants in South Africa [115]. Field testing is needed to see if these responses translate to effective deterrence in the wild. Santiapillai and Read [182] conjectured as to the possibility of masking the odor of ripening crops such as rice to reduce elephant consumption. General unpleasant odor mixtures also are displeasing to elephants also may aid in deterring elephants. For example, Oniba and Robertson [183] used a mixture of chili, garlic, ginger, neem leaves, eggs, and cow or elephant dung, which in this preliminary study was effective at reducing crop raiding. Rasmussen and Riddle [184] tried a mixture of plant and animal products as an olfactory deterrent, and they concluded that olfaction alone was insufficient to deter elephants from food. This is a good recommendation to keep in mind; it is unlikely that a single sensory ‘silver bullet’ will be identified to mitigate crop raiding and similar forms of human-elephant conflict. Nevertheless, chemical signals are vitally important to elephants, so their potential roles for husbandry and conservation should not be underestimated.

## 4. Conclusions

Elephants serve as an excellent model system for mammalian chemical ecology. Behavioral, biochemical, genetic, and physiological studies all point to elephants as highly reliant on their chemical senses. Their presence in zoological facilities, their capability of learning, and the relative ease at which they can be observed in the wild make them good candidates as a model system for further understanding mammalian chemical ecology [185]. African elephants use urinary odors to recognize conspecifics and create expectations of where particular individuals are located spatially [96]. African and Asian elephants can distinguish scents from different humans through operant conditioning [186,187]. Studies like these, the ability to work with and train elephants, and their distinct, readily displayed chemosensory behaviors show that elephants and their use of odors make them good subjects for the investigation of cognition related to olfaction in non-primate mammals.

Despite the social complexity and cognitive capabilities of elephants, single or a small set of compounds may well be important in orchestrating elephant society (e.g., [94] for chemical signaling in mammals). Challenges for the 21st century remain to identify additional pheromones in elephants and to examine further how these signals work in elephant society, especially in a multimodal way with other signal types (e.g., [188]). As Rasmussen and her colleagues have shown, the identification of the pheromone (Z)-7-dodecen-1-yl can be used as a tool to understand pheromone transport, associated physiological conditions, and condition-dependent behavioral responses. We should eagerly pursue identifying the intraspecific chemical signals (i.e., pheromones) and relevant context in elephants and all vertebrates [189]. Further, as illustrated in this review, numerous intraspecific and interspecific interactions are mediated at least in part by chemical signals in elephants. Thus, we should continue to explore the broad chemical ecology of elephants.

## Figures and Tables

**Figure 1 animals-11-02860-f001:**
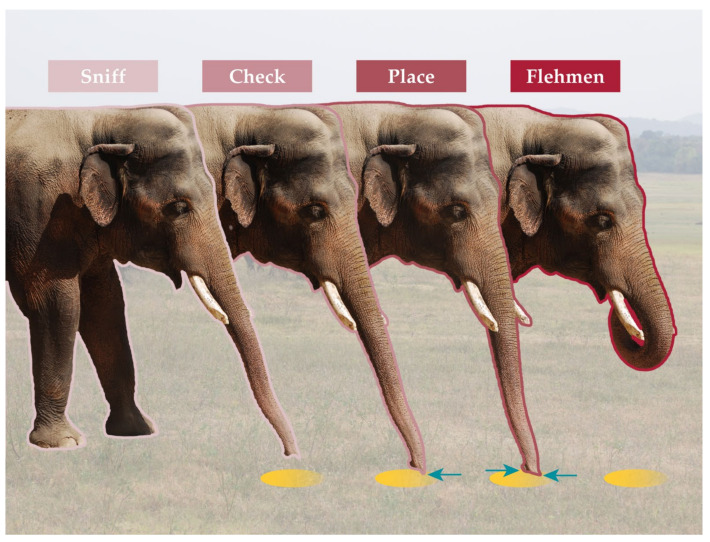
Series of stereotyped chemosensory behaviors present in African and Asian elephants (shown here with a male Asian elephant), adapted from [45,46]. Sniff: elephant holds trunk over signal of interest. Check: dorsal tip (“finger”) of trunk contacts signal. Place: entire end of trunk contacts signal. Flehmen: trunk transports signal to orifice of vomeronasal ducts. In order, these behaviors indicate the level of interest in a signal and may be used readily in bioassays among wild and captive elephant populations. Yellow circles are chemical signals of interest, and blue arrows show points of contact of the trunk with the chemicals.

**Figure 2 animals-11-02860-f002:**
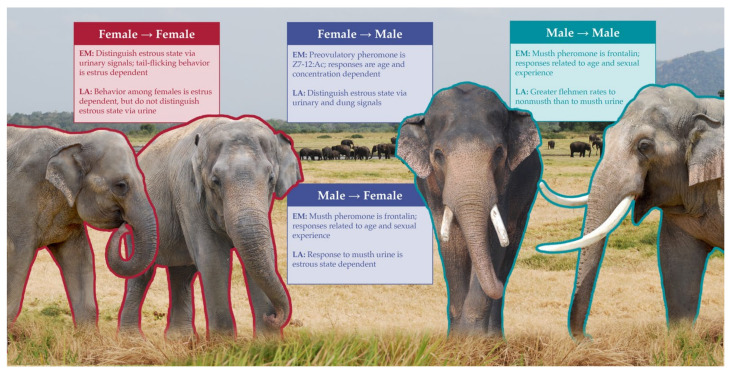
Summary of 21st-century findings in intraspecific chemical signaling among elephants. EM = *Elephas maximus*, the Asian elephant. LA = *Loxodonta africana*, the African savanna elephant. Red indicates female–female signals, blue indicates male–male signals, and purple indicates intersexual signals.

**Figure 3 animals-11-02860-f003:**
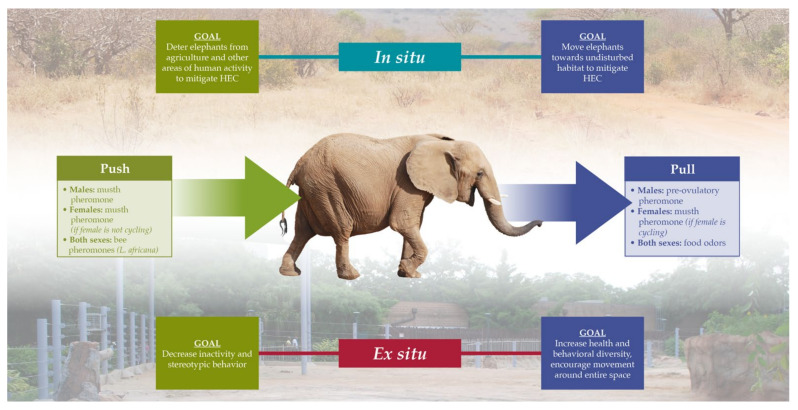
Potential applications of chemical signals for the conservation, management, and husbandry of elephants, using the push-pull approach [142] to manipulate elephant movement.

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
