# Peer review of "The Chemical Ecology of Elephants: 21st Century Additions to Our Understanding and Future Outlooks"

_animals, 2021, doi:10.3390/ani11102860_

Round 1

Reviewer 1 Report

Overall:

I think the authors do a good job compiling African savanna and Asian elephant literature in this review of elephant chemical ecology. This manuscript will be a useful contribution to elephant science, facilitating a big picture and up-to-date understanding of elephant chemical signaling.

My biggest concern is how the paper is structured. The way this manuscript is currently separated into before and after the turn of the 21st century makes specific ideas harder to follow and some sections feel redundant. I would highly recommend sub-sectioning the paper by themes. This would highlight the scientific themes instead of placing focus on when they occurred. For example: the importance of musth is heavily discussed in paragraph five and then after several other topics are introduced the authors return to musts in paragraph 11 with the discovery of frontalin. Discussions of the chemicals involved in musth would be clearer if they were grouped together. Broad sections could include (not limited to) physical anatomy, behavioral responses, pheromones, and inter- and intraspecific signaling/ chemical ecology.

The authors need to consistently refer to other scientists either all with their first & last names (preferably with title) or simply by their last name as is my common in ecology. Currently, male authors (LaDue [line 207]; Oniba [line 397]) are referred to by last names and female authors sometimes have only first or nicknames or both first and last (Bets Rasmussen [lines 199 and 422] or just Bets line 207).

Line specific comments:

Line 57: Change synchronizing breeding to synchronized breeding

Line 177: Are there any comparison studies that show elephants have an unrivaled perception? Both rats and dogs can detect bombs by smell, similarly having no evolutionary context. Without a study to cite I would remove the last phrase: “to an extent perhaps unrivaled by other mammals.”

Line 199: At a minimum add “Dr.” to L.E.L “Bets” Rasmussen, generally, it feels too familiar/unprofessional for a scientific paper. I would also clarify the sentence. As it currently reads, it sounds like Dr. Rasmussens’s biggest contribution is the identification of frontalin, as in her personal best achievement. I’m guessing the authors intended to say: one of the biggest contributions to science is Dr. Rasmussen’s discovery of frontalin. I would recommend rewording to “Perhaps the most profound contribution recently was discovery of the compound frontalin…” This avoids inappropriate lack of formality and lack of clarity.

Lines 270-273: This sentence is too long and is unclear. Please break it into smaller sentences.

Lines 293-300: The discussion of bees is non sequitur to the discussion of humans and predators. I would move this to the foraging paragraph below.

Line 372-374: I would explicitly state that the elephants avoided areas sprayed. As it is currently worded, there is no mention of the results of the research.

Line 387: Remove the dangling prepositional phrase. “In a lab experiment” is unnecessary.

Line 397: Similar to a comment above: was the deterrent successful? Go beyond simply saying Oniba and Robertson used a mixture.

Line 413: How do these studies show that elephants make good subjects? Be more specific.

The last paragraph jumps around. Pheromones seem like only a portion of the all chemical signaling in elephants. The authors should consider finishing on a broader note. The sentence starting on line 419 is more what is expected for a final paragraph.

Line 425: This sentence does not make much sense. Please try to clarify the difference between broad chemical ecology and deciphering chemical language.

Author Response

I think the authors do a good job compiling African savanna and Asian elephant literature in this review of elephant chemical ecology. This manuscript will be a useful contribution to elephant science, facilitating a big picture and up-to-date understanding of elephant chemical signaling.

My biggest concern is how the paper is structured. The way this manuscript is currently separated into before and after the turn of the 21st century makes specific ideas harder to follow and some sections feel redundant. I would highly recommend sub-sectioning the paper by themes. This would highlight the scientific themes instead of placing focus on when they occurred. For example: the importance of musth is heavily discussed in paragraph five and then after several other topics are introduced the authors return to musts in paragraph 11 with the discovery of frontalin. Discussions of the chemicals involved in musth would be clearer if they were grouped together. Broad sections could include (not limited to) physical anatomy, behavioral responses, pheromones, and inter- and intraspecific signaling/ chemical ecology.

We appreciate the organizational comment. None of the other three reviewers had an issue with the organization. We organized in this fashion because as we state in the manuscript, several reviews on elephant chemical ecology were published in the late 1990s. We did not wish to go back over the ground covered in these reviews, so we gave short summaries of what was known at that time. Our logic was to give a holistic overview up to the start of the 21st Century and then develop lines of research that have occurred in the past 20 years. We do not disagree that the reviewer offers an alternative means to organize the review; it simply is not the organization that we felt was most salient given the charge by the special issue organizer to provide a review on recent findings in elephant chemical ecology.

The authors need to consistently refer to other scientists either all with their first & last names (preferably with title) or simply by their last name as is my common in ecology. Currently, male authors (LaDue [line 207]; Oniba [line 397]) are referred to by last names and female authors sometimes have only first or nicknames or both first and last (Bets Rasmussen [lines 199 and 422] or just Bets line 207).

Changed to last name. See also our reply to the Line 199 suggested edits.

Line specific comments:

Line 57: Change synchronizing breeding to synchronized breeding: Done

Line 177: Are there any comparison studies that show elephants have an unrivaled perception? Both rats and dogs can detect bombs by smell, similarly having no evolutionary context. Without a study to cite I would remove the last phrase: “to an extent perhaps unrivaled by other mammals.”

Our statement was based on the Niimura et al. 2014 genetic study showing elephants had the largest number of genes for olfactory receptors of those animals tested to date. The study included dogs and rats. We used ‘perhaps’ to indicate that this was hypothetical not fact. However, to avoid confusion, we removed the statement.

Line 199: At a minimum add “Dr.” to L.E.L “Bets” Rasmussen, generally, it feels too familiar/unprofessional for a scientific paper.  I would also clarify the sentence. As it currently reads, it sounds like Dr. Rasmussens’s biggest contribution is the identification of frontalin, as in her personal best achievement. I’m guessing the authors intended to say: one of the biggest contributions to science is Dr. Rasmussen’s discovery of frontalin. I would recommend rewording to “Perhaps the most profound contribution recently was discovery of the compound frontalin…” This avoids inappropriate lack of formality and lack of clarity.

We made the suggested changes then added a note in line 208 to explain why the investigation was not completed, [Using Rasmussen here in place of Bets would not work because she is no longer introduced by name in L199]. We used Dr. L.E.L. Rasmussen in the note to maintain formality. Thank you for your comment.

Lines 270-273: This sentence is too long and is unclear. Please break it into smaller sentences.

We placed a period after “anatomy” and revised the final statement to make it a complete sentence. In response to comments from another reviewer, we added to this paragraph as well.

Lines 293-300: The discussion of bees is non sequitur to the discussion of humans and predators. I would move this to the foraging paragraph below.

Done, moved to next paragraph.

Line 372-374: I would explicitly state that the elephants avoided areas sprayed. As it is currently worded, there is no mention of the results of the research.

The point of these lines was to introduce the matrix SPLAT not to discuss the results of the bee pheromone study. We reworded to clarify.

Line 387: Remove the dangling prepositional phrase. “In a lab experiment” is unnecessary. Done.

Line 397: Similar to a comment above: was the deterrent successful? Go beyond simply saying Oniba and Robertson used a mixture. Done.

Line 413: How do these studies show that elephants make good subjects? Be more specific.

Text added as follows: “Studies like these, the ability to work with and train elephants, and their distinct, readily displayed chemosensory behaviors show that elephants and their use of odors make them good subjects for the investigation of cognition related to olfaction in non-primate mammals.”

The last paragraph jumps around. Pheromones seem like only a portion of the all chemical signaling in elephants. The authors should consider finishing on a broader note. The sentence starting on line 419 is more what is expected for a final paragraph.

We appreciate this perspective and have reorganized the final paragraph to emphasize the general chemical ecology of elephants.

Line 425: This sentence does not make much sense. Please try to clarify the difference between broad chemical ecology and deciphering chemical language. Done

We appreciate the input from Reviewer 1. Please note that some line numbers have changed slightly because of edits.

Reviewer 2 Report

Overall this is a fine review of the current state of knowledge in this field. There is a large number of references but very little variability, presumably due to the small cadre of scientist that work in this specific field. The manuscript has merit and should serve as a good reference for the work done so far, there are some minor changes I would recommend, and some clarifications to logic that should be made prior to publication.

Beginning on line 25… This sentence and the following are wordy and do not to be as complicated. Consider rewrite.

The sentence beginning on line 55 (“While musth…) needs a citation for the ansychronicity and inter-cycle period for females.

The sentence beginning on line 180 (Interesting, despite…) is confusing and eludes to the fact that other mammals have an accessory olfactory bulb, without example of citation.

The sentence in line 345, comes across as an opinion (“…They are likely to be resistant to habituation”) but you make the correlation between biological relevance and habituation with no examples of that existing. Is that a commonly held belief or is it validated? If it is the latter it should be cited.  Specifically because you go on to mention other biologically relevant chemical signals (capsaicin, etc…) later in the paragraph, which in practice often lead to habituation.

Author Response

Overall this is a fine review of the current state of knowledge in this field. There is a large number of references but very little variability, presumably due to the small cadre of scientist that work in this specific field. The manuscript has merit and should serve as a good reference for the work done so far, there are some minor changes I would recommend, and some clarifications to logic that should be made prior to publication.

Beginning on line 25… This sentence and the following are wordy and do not to be as complicated. Consider rewrite.

We modified the second sentence (the first is unchanged, see below).

Chemical signals are the oldest and most ubiquitous means of mediating intra- and interspecific interactions. The three extant species of elephants, the Asian elephant and the two African species, savanna and forest, share sociobiological patterns in which chemical signals play a vital role.

The sentence beginning on line 55 (“While musth…) needs a citation for the ansychronicity and inter-cycle period for females.

Done.

The sentence beginning on line 180 (Interesting, despite…) is confusing and eludes to the fact that other mammals have an accessory olfactory bulb, without example of citation.

We have reworded and added citations.

The sentence in line 345, comes across as an opinion (“…They are likely to be resistant to habituation”) but you make the correlation between biological relevance and habituation with no examples of that existing. Is that a commonly held belief or is it validated? If it is the latter it should be cited. We do not know of this statement being validated in elephants; therefore, we modified our wording making this statement a suggestion.  Specifically because you go on to mention other biologically relevant chemical signals (capsaicin, etc…) later in the paragraph, which in practice often lead to habituation.

To clarify, compounds like capsaicin are chemicals that may result in a response when detected by elephants. We are not saying that these are evolved means of information transfer (a ‘signal’).

We appreciate the inputs from Reviewer 2. Please note that some line numbers have changed slightly because of edits.

Reviewer 3 Report

Excellent review of an area that has not been extensively studied.  While a review, it points out areas of chemical ecology that are available for future investigation.  Well written and organized. 

Author Response

Thank you. We appreciate your review.

Reviewer 4 Report

This is a well written review on how the study of chemical signaling in  ex situ and in situ populations of Asian and African elephants advances by time and the usefulness of pheromones for conservation of wild elephant  population and management of elephants in captivity. The references are valuable contribution to those who wish to study chemical signaling in elephants. Allow me to add a minor comment. Loads of study have shown that pheromones are traceable in urine, faeces, saliva, vaginal mucus, milk, sweat and scent glands of vertebrates and they are used for communication between animals of same species for the facilitation of mating (swine) and nestling  (rodents) behaviours as well as mother-infant recognition/interaction (humans), fight or flight responses (insects). No studies have yet done whether elephant saliva contains pheromones. It would make a worthwhile contribution if the authors add some lines on their thoughts on whether saliva  can be a source of pheromones in elephants.

Author Response

Re: possible chemical signals in saliva

We have added text and citations to address this omission. We appreciate the comments by Reviewer 4.